# MODEL-INVARIANT STATE ABSTRACTIONS FOR MODEL-BASED REINFORCEMENT LEARNING

## ABSTRACT

Accuracy and generalization of dynamics models is key to the success of model-based reinforcement learning (MBRL). As the complexity of tasks increases, learning accurate dynamics models becomes increasingly sample inefficient. However, many complex tasks also exhibit sparsity in dynamics, i.e., actions only have a local effect on the system dynamics. In this paper, we exploit this property with a causal invariance perspective in the single-task setting, introducing a new type of state abstraction called *model-invariance*. Unlike previous forms of state abstractions, model-invariant state abstraction leverages causal sparsity over state variables. This allows for compositional generalization to unseen states, something that non-factored forms of state abstractions cannot do. We prove that an optimal policy can be learned over exact model-invariance state abstraction and show improved generalization in a simple toy domain. Next, we propose a practical method to approximately learn a model-invariant representation for complex domains and validate our approach by showing improved modelling performance over standard maximum likelihood approaches on challenging tasks, such as the MuJoCo-based Humanoid. Finally, within the MBRL setting we show strong performance gains with respect to sample efficiency across a host of continuous control tasks.

## 1 INTRODUCTION

Model-based reinforcement learning or MBRL [4, 15] is a popular framework for data-efficient learning of control policies. At the core of MBRL is learning an environmental dynamics model and using it to: 1) fully plan [14, 11], 2) augment the data used by a model-free solver [51, 54], or 3) use as an auxiliary task while training [35, 58]. However, learning a dynamics model — similar to other supervised learning problems — suffers from the issue of generalization, since the data we train on is not necessarily the data we test on. This is a persistent issue that is worsened in MBRL as even a small inaccuracy in the dynamics model or changes in the control policy can result in visiting completely unexplored parts of the state space [1]. This advocates for the need to learn models capable of generalizing well. Various workarounds for this issue have been explored in the past; for example, combining local but simple models with global, more expressive models [36, 22], using an ensemble of models to handle uncertainty in estimates [11, 33] and coupling the model and policy learning processes [34] so that the model is always accurate to a certain threshold. However, these approaches do not stem from a representation learning viewpoint and thus fail to leverage special structure in dynamics for better generalization.

This paper studies how to improve generalization capabilities through careful state abstraction. In particular, we leverage two existing concepts to define a new kind of state abstraction. The first concept is that many real world problems exhibit *sparsity* in the local dynamics — given a set of state variables, each variable at timestep $t + 1$ only depends on a small subset (i.e., local) of all variables in the previous timestep $t$ (see Figure 1). The second concept is the principle of causal invariance, which dictates that given a set of candidate features, we should aim to build representations that comprise *only* those features that are consistently necessary for predicting the target variable of interest across different interventions [46]. In the MBRL context, we can cast model learning with causal invariance as a supervised objective where the target variables are the next state variables and input features are the current state and action variables (the probable set of causal predictors of the target).

Intuitively, since we learn a predictor that only takes into account invariant features that consistently predict the target variable well, it is likely to contain the true causal features, and therefore, will generalize well to all possible shifts in the data distribution. The two concepts of sparsity and causality are intertwined in that they both are forms of inductive biases that surround the agent dynamics [23]. This paper shows how causal invariance tools can be effectively used to exploit sparsity in dynamics, leading to improved model generalization. Given basic exploratory assumptions, we analyze this question theoretically and show empirically that we can learn a model that generalizes well on state distributions induced by policies distinct from the ones used while learning it. To do this, we introduce a new state abstraction, model-invariance, which leverages sparsity over state variables. We connect this abstraction viewpoint to a concrete problem of generalization in model-based RL, that of

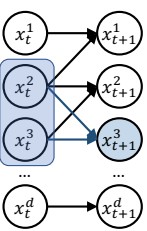

Figure 1: **Graphical model of sparsity** across state variables. The state variable $x^3_{t+1}$ (shaded in blue) only depends on two variables $x^3_t$ and $x^2_t$ (in the blue box). Predicting $x^3_{t+1}$ using the other variables (say $x^1_{t+1}$) can result in spurious correlations which lead to poor generalization.

arising spurious correlations in the dynamics model, even when all state variables are task relevant. Having built enough intuition, we then introduce a practical method that approximates learning a model-invariant representation for more complex domains that use function approximation (i.e., neural networks (NNs)). We empirically observe that model invariance leads to better model generalization for domains such as the MuJoCo-based Humanoid. Our method is simple to implement and shows strong performance on multiple MuJoCo tasks, outperforming state of the art model-based methods in the low-data regime.

## 2 PRELIMINARIES

**RL Setup**. We consider the agent's interaction with the environment as a discrete time $\gamma$-discounted Markov Decision Process (MDP) [47] $\mathcal{M} = (\mathcal{X}, \mathcal{A}, P, R, \gamma, \mu_0)$, where $\mathcal{X} \subseteq \mathbb{R}^d$ is a finite but arbitrarily large state space and $\mathcal{A}$ is the and action space; $P \equiv P(x'|x, a)$ is the transition kernel; $R \equiv r(x, a)$ is the reward function with the maximum value of $R_{\max}$; $\gamma \in [0, 1)$ is the discount factor; and $\mu_0$ is the initial state distribution. Let $\pi : \mathcal{X} \to \Delta_{\mathcal{A}}$ be a stationary memoryless policy, where $\Delta_{\mathcal{A}}$ is the set of probability distributions on $\mathcal{A}$. The value function of a policy $\pi$ at a state $x \in \mathcal{X}$ is defined as $V^\pi(x) \equiv \mathbb{E}[\sum_{t \geq 0} \gamma^t r(x_t, a_t)|x_0 = x, \pi]$. Similarly, the action-value function of $\pi$ is defined as $Q^\pi(x, a) = \mathbb{E}[\sum_{t \geq 0} \gamma^t r(x_t, a_t)|x_0 = x, a_0 = a, \pi]$. The Bellman optimality operator $\mathcal{T} : \mathbb{R}^{|\mathcal{X} \times \mathcal{A}|} \to \mathbb{R}^{|\mathcal{X} \times \mathcal{A}|}$ is defined as $\mathcal{T}Q(x, a) = r(x, a) + \gamma \langle P(\cdot|x, a), \max_{a'} Q(\cdot, a') \rangle$. In this work, we assume that all $d$ state variables $x^1, x^2, ..., x^d$ are useful for the task in hand, i.e., we are given the full state. Furthermore, we assume that the transition dynamics over the full state are factorized. More formally:

**Assumption 1.** *(Transition Factorization) For given full state vectors $x_t, x_{t+1} \in \mathcal{X}$, action $a \in \mathcal{A}$, and $x^i$ denoting the $i^{th}$ dimension of state $x$ we have $P(x_{t+1}|x_t, a) = \prod_i P(x^i_{t+1}|x_t, a)$.*

Note that this is a weaker assumption than factored MDPs [31, 24] as we do not assume a corresponding factorization of the reward function.

**Invariant Causal Prediction**. Invariant causal prediction (ICP) [46] considers the problem of learning an invariant representation w.r.t. spurious correlations that arise due to noise in the underlying causal model (unknown) describing a given system. The key observation is that if one considers the direct causal parents of a response/target variable of interest ($Y$), then the conditional distribution of $Y$ given these direct causes **PA**($Y$) does not change across interventions on any variable except $Y$. Therefore, ICP suggests collecting data into different environments (corresponding to different interventions), and to output the set of variables $X_i$ for which a learned predictor of $Y$ remains the same given $X_i$ across the multiple environments with high probability.

**State Abstractions**. State abstractions allow us to map behaviorally-equivalent states into a single abstract state, thus simplifying the learning problem, which then makes use of the (potentially much smaller set of) abstract states instead of the original states [5]. In principle, any function approximation architecture can act as an abstraction, since it attempts to group similar states together. Therefore, exploring the properties of a representation learning scheme as a state abstraction help

develop stronger intuition for building practical algorithms. In the next section, we build our theory based on this connection. A well known state abstraction is bisimulation [19, 48, 37]. Formally, an abstraction $\phi : \mathcal{X} \mapsto \mathcal{S}$ is a bisimulation if for any two states $x_1, x_2$ and next state $x \in \mathcal{X}$, abstract state $s \in \mathcal{S}$, $a \in \mathcal{A}$ where $\phi(x_1) = \phi(x_2)$, we have $R(x_1, a) = R(x_2, a)$, and $\sum_{x \in \phi^{-1}(s)} P(x|x_1, a) = \sum_{x \in \phi^{-1}(s)} P(x|x_2, a)$. Since an exact equivalence is not practical, prior work deals with approximate variants through the notion of $\epsilon$-closeness [29].

## 3  MODEL INVARIANCE AND ABSTRACTIONS

Current state abstractions such as bisimulation do not provide support for sparse structures. Instead, closeness in abstract states is defined based on probabilities of all state variables together, i.e., $x$. To remedy this, we introduce a new state abstraction, called model-invariance, which is specific to each state variable $x^i$. Formally, an invariant abstraction $\phi_i$ is one which has the same transition probability of the $i^{\text{th}}$ next state variable $x^i$ for any two given states $x_1$ and $x_2$, i.e., $P(x^i|x_1, a) = P(x^i|x_2, a)$. Note that if we assume factored rewards, we can define a corresponding reward-based invariant abstraction that parallels the bisimulation abstraction more closely, but we focus here on the reward-free setting. Since it is impractical to ensure this equivalence exactly, we can use an approximate definition which ensures an $\epsilon$-closeness.

**Definition 1.** *(Approximate Model Invariance)* $\phi$ *is* $\epsilon_{i,P}$-*model-invariant if for each index* $i$,

$$\sup_{a \in \mathcal{A}, x, x_1, x_2 \in \mathcal{X}, \phi(x_1):=\phi(x_2)} \left\| P(x^i|x_1, a) - P(x^i|x_2, a) \right\| \leq \epsilon_{i,P}.$$

$\phi$ *is* $\epsilon_R$-*model-invariant if* $\epsilon_R = \sup_{a \in \mathcal{A}, x_1, x_2 \in \mathcal{X}, \phi(x_1)=\phi(x_2)} \left| R(x_1, a) - R(x_2, a) \right|$.

Based on the definition, it intuitively seems like applying the abstraction over the original transition distribution should be close to the transition distribution over abstract states. This can be precisely written in the following lemma, which is a necessary result in ensuring that an optimal policy can still be learnt over a model-invariant abstraction:

**Lemma 1.** *(Model Error Bound) Let* $\phi$ *be an* $\epsilon_{i,P}$-*approximate model-invariant abstraction on MDP* $M$. *Given any distributions* $p_{s_i} : s_i \in \phi_i(\mathcal{X})$ *where* $p_{s_i}$ *is supported on* $\phi^{-1}(s_i)$ *and* $p_s = \prod_{i=1}^{d} p_{s_i}$, *we define* $M_\phi = (\phi(\mathcal{X}), \mathcal{A}, P_\phi, R_\phi, \gamma)$ *where* $P_\phi(s, a) = \mathbb{E}_{x \sim p_s} \left[ P(\cdot|x, a) \right]$ *and* $R_\phi = \mathbb{E}_{x \sim p_s} \left[ R(x, a) \right]$. *Then for any* $x \in \mathcal{X}$, $a \in \mathcal{A}$,

$$\| P_\phi(s, a) - \Phi P(x, a) \| \leq \sum_{i=1}^{d} \epsilon_{i,P},$$

where $\Phi P$ denotes the *lifted* version of $P$, where we take the next-step transition distribution from observation space $\mathcal{X}$ and lift it to latent space $\mathcal{S}$ (Proof in Appendix 2). $P_\phi$ refers to the transition probability of a MDP that acts on the states $\Phi(\mathcal{X})$, rather than the original MDP. Finally, note that we are particularly concerned with the case where each $x^i$ is atomic in nature, i.e., it is not further divisible. Such a property ensures that model-invariance does not collapse to bisimulation[1].

## 4  CAUSAL INVARIANCE IN MODEL LEARNING

Having defined model-invariant abstractions, we are now ready to provide connections between causal invariance and model learning in RL. For simplicity, we can assume that there exists a linear structural equation model [45] that consists of the $d$ state variables and action $a$ as the features $X$ and the next state variable $x_{t+1}^i$ as the target $Y$, for each index $i$. Similar to the ICP setting(Section 2), we can define the different environments as follows:

**Assumption 2.** *(ICP Environments) For each* $e \in \mathcal{E}$: *the experimental setting* $e$ *arises due to one or several interventions on variables from* $(x_t^1, ..., x_t^d, a_t)$ *but not on* $x_{t+1}^i$; *here, we allow for do-interventions [44] or soft-interventions [18].*

---

[1]In the simplest case index $i$ describes each state dimension, i.e., being atomic. However in general the index could be a grouping of different dimensions as well. Consider the case where there is only one index over which we build a model invariance abstraction. Now $x^1$ would then correspond to the entire state $x$. Since Definition 1 would be satisfied trivially, we do not gain anything on the sparsity level, thus collapsing to bisimulation.

For our purposes, each intervention corresponds to a change in the current state and action distribution. This change in distribution can be realized as a change in the policy. Therefore, each policy $\pi$ defines an ICP environment $e$. It can be shown that under Assumption 2 the direct causes, i.e., parents of $x_{t+1}^i$, define a valid support over invariant predictors, namely $S^* = \mathbf{PA}(x_{t+1}^i)$. [2]. The key idea therefore is to make sure that in predicting each next state variable $x_{t+1}^i$ we use only its set of invariant predictors and not all state variables and actions (see Figure 1). With this intuition, it becomes clearer why our original model learning problem is inherently tied with learning better representations, in that having access to a representation that discards excess information for each state variable (more formally, a causally invariant representation), would be more robust to spurious correlations and thus, at least in principle, lead to improved generalization performance across different parts of the state space. In fact, such a causally invariant representation obeys the properties of a model-invariant abstraction. Formally,

**Proposition 1.** *For the abstraction $\phi_i(x) = [x]_{S_i}$, where $S_i = \mathbf{PA}(x_{t+1}^i)$, $\phi_i$ is model-invariant.*

Proof is provided in Appendix 2. It now becomes easy to see that sparsity in dynamics is central to what we have discussed so far, since if we do not have sparsity, the causally invariant representation trivially reduces to the original state $x$, thus resulting in no state aggregation. However, when sparsity is present, following the above definition leads to a representation $\phi$ such that any two components $\phi_i$ and $\phi_j$ are decorrelated, since each encodes information about a different subset of state variables. Finally, we show that learning a transition model over a model-invariant abstraction $\phi$ and then planning over this model is optimal.

**Theorem 1.** *(Value bound) If $\phi$ is an $\epsilon_R, \epsilon_{i,P}$ approximate model-invariant abstraction on MDP $M$, and $M_\phi$ is the abstract MDP formed using $\phi$, then we can bound the loss in the optimal state action value function in both the MDPs as:*

$$\left\| [Q_{M_\phi}^*]_M - Q_M^* \right\|_{2,\nu} \le \frac{\sqrt{C}}{1-\gamma} \left\| [Q_{M_\phi}^*]_M - \mathcal{T}[Q_{M_\phi}^*]_M \right\|_{2,\mu},$$

$$\left\| [Q_{M_\phi}^*]_M - \mathcal{T}[Q_{M_\phi}^*]_M \right\|_{2,\mu} \le \epsilon_R + \gamma \Big( \sum_{i=1}^d \epsilon_{i,P} \Big) \frac{R_{max}}{2(1-\gamma)},$$

where $C$ refers to the concentrability coefficient as defined in [9]. Here, an admissible distribution $\nu$ refers to any distribution that can be realized in the given MDP by following a policy for some timesteps, while $\mu$ refers to the distribution from which the data is generated. Proof and all details surrounding the theoretical results are provided in Appendix 2. This result is important because we can follow this with standard sample complexity arguments that will have a logarithmic in $|\phi|$ dependence, thus guaranteeing that learning in this abstract MDP is faster and only incurs the above described sub-optimality.

So far, we have embedded the RL model learning problem in the ICP framework. This highlights the connection between generalization issues in MBRL and arising spurious correlations in identifying the true causal predictors of each state variable. Furthermore, we have explored the state abstraction properties of the causally invariant representations which avoid spurious correlations. In the next section, we now show how ICP can be used as a sub-routine in learning a more generalized model in the simple setting of a linear MDP.

## 5 LINEAR CASE: CERTAINTY EQUIVALENCE

In the simpler setting of tabular RL, estimating the model using transition samples and then planning over the learned model is referred to as certainty equivalence [6]. Particularly for estimating the transition model, it considers the case where we are provided with $n$ transition samples per state-action pair, $(x_t, a_t)$ in the dataset $D_{x,a}$, and estimate the model as

$$P\big(x_{t+1}|x_t, a_t\big) = \frac{1}{n} \sum_{\bar{x} \in D_{x,a}} \mathbb{I}(\bar{x} = x_{t+1}).$$

---

[2] The proof follows directly by applying Proposition 1 of Peters et al. [46] (which itself follows from construction) to each state variable indexed by dimension $i$

Now, when next state components do not depend on each other given the previous state and action (i.e., Assumption 1), we can re-write $P(x_{t+1}|x_t, a_t)$ as $\prod_i P(x_{t+1}^i|x_t, a_t)$. Furthermore, if we know the causal parents of $x_{t+1}^i$, we can instead empirically estimate the true transition probabilities as

$$P(x_{t+1}^i|x_t, a_t) = P(x_{t+1}^i|\mathbf{PA}(x_{t+1}^i), a_t) = \frac{1}{nk} \sum_{\bar{x} \in D} \mathbb{I}(\bar{x} = x_{t+1}^i), \quad (1)$$

where $D = \bigcup_{i=1}^{k} D_{x,a}, \ x \in \phi_i^{-1}(\bar{x})$. From Proposition 1, we know that such an invariant solution exists, and is defined by the causal parents of each state variable. Therefore, in the linear dynamics case, given data from multiple environments (different policies), we can use ICP to learn the causal parents of each state variable and then estimate the probability of a certain transition using Eq. 1[3]. We refer to this as the invariant model learner and detail the procedure in Algorithm 1. On the other hand, if we do extract the causal parent variables and instead use all state variables in Eq. 1, we would get the standard maximum likelihood (MLE) learner.

---

**Algorithm 1** Linear Model-Invariant MBRL

1: **Input** Replay buffer $\mathcal{D}$ containing data from multiple policies/envs, confidence parameter $\alpha$;
2: **for** state variable $i = 1, \ldots, d$ **do**
3:   $\mathcal{S}_i \leftarrow \text{ICP}(i, \mathcal{D}, \alpha)$; (Appendix **??**)
4:   Estimate $\hat{P}_i$ from $\mathcal{D}$ using Eq. 1;
5: **end for**
6: Estimate transition probability kernel $\hat{P} \leftarrow \Pi_i \hat{P}_i$
7: **for** state $s_j, \ j = 1, \ldots, N$ **do**
8:   $\pi_R(s_j) \leftarrow \text{Plan}(R, \hat{P}, s_j)$
9: **end for**

---

The invariance based solution avoids spurious correlations more than the MLE learner, resulting in better generalization. To see this, consider a simple linear MDP with three state variables $(x^1, x^2, x^3)$, each depending only on its own $(\mathbf{PA}(x_{t+1}^i) = x_t^i)$, and taking integer values between $[-10, 10]$. The exact details of the MDP are described in Appendix 3.1. We consider three different distributions corresponding to three different policies, each describing an ICP environment. In Table 1, we compare our invariant learner with a standard MLE learner and show how their error with the true probability of transition varies as the number of samples grows.

| Samples | MLE | Model-Invariance |
|---|---|---|
| 100 | - $\pm$ - | **9.3** $\pm$ 1.3 |
| 200 | 24.6 $\pm$ 9.23 | **5.7** $\pm$ 1.5 |
| 500 | 14.5 $\pm$ 2.75 | **2.7** $\pm$ 0.7 |
| 2000 | 9.6 $\pm$ 1.72 | **1.8** $\pm$ 0.3 |
| 5000 | 6.4 $\pm$ 1.46 | **1.6** $\pm$ 0.3 |

Table 1: **Linear MDP Transition Prediction Error**. Consider the simple linear MDP from Appendix 3.1. We compare the estimated transition probability of our invariant learner with the MLE learner (lower is better). The invariant learner converges faster and more stably to the true solution. Mean and std. err. over 15 random seeds. Order of magnitude of the errors is 1e-3.

Note that Table 1 represents the results for one environment, specified by a fixed policy that is used for data collection. If the policy changes, it would result in a different environment as described in Section 3. Our ideal scenario is to find a predictive model that is closest to the true model for all environments. We find that the invariant learner quickly converges to approximately the same solution across all training environments, with just a few data samples. On the other hand, the standard MLE learner results in different solutions for each training environment in the low data regime. The solution provided at test time in such a case is an average of all such solutions found during training, which is clearly off the true probability (higher error in Table 1).

It is worth noting that this example assumes linearity in dynamics, which allows us to use the ICP procedure from Peters et al. [46]. In the general non-linear case, this is not possible. To that end, in the next section we will describe a practical method that leverages ideas from self-supervised learning to exploit sparsity in an end-to-end manner.

## 6   NON-LINEAR CASE: LEARNING PRACTICAL MODEL-INVARIANT REPRESENTATIONS

We now introduce a practical algorithm for learning model-invariant representations where we relax the following assumptions: linearity in dynamics, having access to data from multiple environments,

---

[3]Using Theorem 1, we know that performing planning over this estimated dynamics model would be $\epsilon$-optimal.

**Algorithm 2** Non-linear Model-Invariant MBRL

1: **Input** Replay buffer $\mathcal{D} = \emptyset$; Value and policy network parameters $\theta_Q, \theta_\pi$, model parameters $\theta_r$, $\theta_f$ for any MBRL algorithm;
2: **for** environment steps $t = 1, \ldots, T$ **do**
3:     Take action $a_t \sim \pi(\cdot|x_t)$, observe $r_t$ and $x_{t+1}$, and add to the replay buffer $\mathcal{D}$;
4:     **for** $M_{\text{model-free}}$ updates **do**
5:         Sample batch $\{x_j, a_j, r_j, x_{j+1}\}_{j=1}^N$ from $\mathcal{D}$;
6:         Run gradient update for the model free components of the algorithm (e.g., $\theta_\pi, \theta_Q$ etc.);
7:     **end for**
8:     **for** $M_{\text{model}}$ updates **do**
9:         Sample batch $\{x_j, a_j, r_j, x_{j+1}\}_{j=1}^N$ from $\mathcal{D}$;
10:       Update reward model ($\theta_r$);
11:       Update invariant dynamics model: $\theta_f \leftarrow$ invariant_update($\theta_f, \nabla_{\theta_f} L_f$) (Appendix 3.2 or Eq. 2);
12:     **end for**
13: **end for**

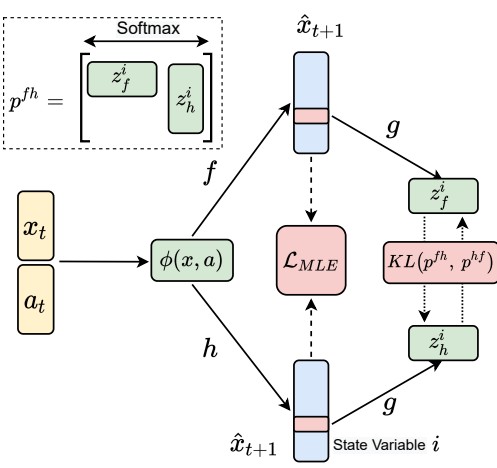

Figure 2: **Architecture for learning model-invariant representations**. Model-invariance uses two (or more) transition dynamics models, denoted by $f$ and $h$, over a common representation $\phi$. The critic $g$ provides a score for a chosen dimension/state variable of the output of $f$ and $h$ models.

and being in the strictly batch setting. Having connected the abstraction viewpoint to the problem of spurious correlations arising in dynamics models, we wish to come up with NN representations that abstract away irrelevant information on a *per-state-variable* level (i.e., $x_t^i$, not $x_t$) and learn them in an end-to-end manner. To that end, we view the task of learning such representations as a self-supervised objective where we want to be invariant to models that exhibit spurious correlations. Consider two (or more) randomly initialized and independently trained (on different samples) dynamics models attached to a common representation $\phi$. We can view these models as augmented versions of the *true* dynamics models since each model captures different spurious correlations. We must be invariant to augmentations in the model space on a *per-state-variable* level (since spurious correlations arise for individual state variable dynamics).

To instantiate this idea, we take inspiration from a recent method called ReLIC [40], which uses a contrastive loss [41] over augmented views of the same data sample and an invariance loss to enforce consistency. ReLIC was shown to be closely connected to doing causal interventions over input variables and enforcing invariance over certain functions (like data augmentations). Here we are exploiting the same connection, with additional modifications. Instead of using augmentations over inputs, we induce augmentations over *models* and then define the invariance loss for a particular dimension — the goal is to eliminate the contribution of spurious features when predicting a particular state variable. Note that doing exact causal interventions is not possible in the large state-action spaces in RL but we can still make strong connections to the causal literature and also use it in simpler settings, as we do in Section 5 (linear case) where we deploy exact invariance tools.

After randomizing two (or more) identical models at the start of training, a model is sampled randomly and is used for minimizing the standard MLE model predictive loss at each optimization step. Simultaneously, an invariance loss defined over the predictions of both models augments the MLE loss. The invariance loss enforces consistency in the prediction of both (all) models by *minimizing the difference in similarities between the prediction of one model w.r.t. the prediction of the other model and vice versa* (Eq. 2). This similarity is computed for a single (randomly selected) state variable at each optimization step, with the specifics being borrowed from the ReLIC objective Mitrovic et al. [40]. Finally, further consistency is encouraged because all models have a common representation with a a bottleneck structure. Our implementation is detailed in Figure 2 and as pseudocode in Appendix 3.2. Although this method is designed to be robust to spurious correlations in dynamics models, in our experiments we will show that the representations learnt for each index $i$, $\phi_i$ actually end up being sparse, a property that motivated the model invariance abstraction back in Definition 1. The overall model invariance loss is:

$$\mathcal{L}_f = \mathbb{E}_{x \sim \mathcal{D}}\left[ \underbrace{\left(f(x_t, a_t) - x_{t+1}\right)^2}_{\text{MLE Loss}} + \underbrace{\text{KL}\left(p^{fh}, p^{hf}\right)}_{\text{Invariance Loss}} \right], \qquad p^{fh} = \text{softmax}\left(\psi^i(f, h)\right), \quad (2)$$

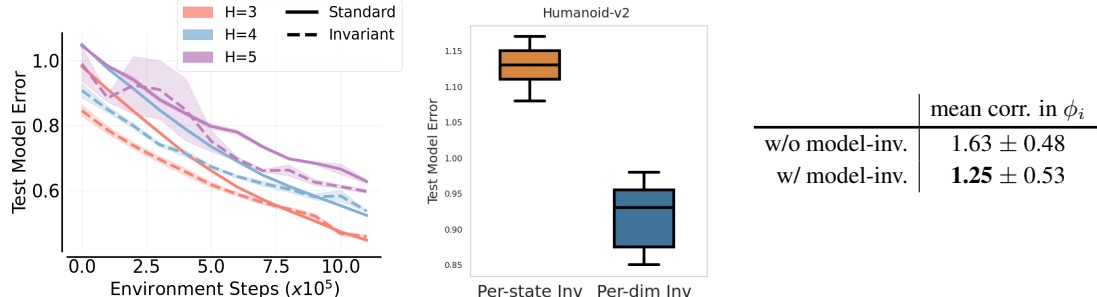

Figure 4: **Invariant Model Learning on Humanoid-v2**. **Left plot**: test model learning error for different horizon values. Mean and std. err. over 10 random seeds. **Middle plot**: modelling error when we enforce per-state vs per-dim invariance. **Table**: mean correlation metric; order of magnitude is $10^3$.

where $\psi^i(f, h) = \Big\langle g\big(f^i(x_t, a_t)\big),\ g\big(h^i(x_t, a_t)\big)\Big\rangle$ is the cosine similarity between the predictions for the models $f$ and $h$ for the state variable indexed by $i$. The output of the inner product is normalized for each sample and thus the resulting vector $g\big(f^i(x_t, a_t)\big)$ is a probability distribution. In our implementation, the function $g$ is a fully connected neural network and is used to project the outputs of the models $f$ and $h$. It is often called the critic network in self-supervised learning losses [10]. Note that the matrix $\psi^i$ is not symmetric since the values at any two symmetric indices are different (they are computed for different samples) — the KL loss remains well-defined.

We will use the invariant model learner described above within a model-based RL algorithm and compare its policy performance to a standard MLE based model learner. A general framework that uses an invariant model learner is outlined in Algorithm 2. For the purposes of this paper, we employ a simple actor-critic setup where the model is used to compute multi-step estimates of the $Q$ value used by the actor learner. A specific instantiation of this idea of model value expansion is the SAC-SVG algorithm proposed in Amos et al. [2] (see Appendix 3.3 for details). It is important to note that the proposed version of model-invariance can be used in combination with any MBRL method and with any type of dynamics model architecture, such as ensembles or recurrent architectures.

## 7 EXPERIMENTS

**Presence of Spurious Correlations**. We first test the presence of spurious correlations in the Humanoid-v2 [53] task by choosing to predict a single dimension (the knee joint) in two contrasting settings; **1) No_Mask:** when all state and action variables are provided as input and, **2) Mask:** when the state variables that are likely uncorrelated to the knee joint are masked (see Appendix 3.5). Having trained different models for the two settings, we observe that **1) No_Mask:** performs worse than **2) Mask:**, for both horizon values in $\{3, 5\}$ (see Figure 3). This verifies that there indeed is an invariant, causal set of parents among the state dimensions and that there could be some interference due to spurious correlations in 1), and thus, it performs worse than case 2). Furthermore, when the dimensions that are likely to be useful in predicting the knee joint are masked,

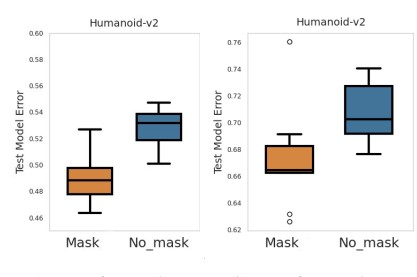

(a) Horizon=3    (b) Horizon=5

Figure 3: **Effect of spurious correlation** on model learning test loss of Humanoid-v2 for a single dimension (knee joint) with two settings: **Mask** and **No_mask**. 10 seeds, 1 std. dev. shaded. Y-axis magnitude order is 1e-3.

then the model error is the highest (worst; not shown in figure). Note that since all variables are necessary for the task in hand, simply discarding some of them is a not a solution to the spurious correlation problem.

**Invariant Model Learning on Humanoid-v2**. We compare the invariant model learner to a standard MLE learner for the Humanoid-v2 task. To observe the effect of the invariance loss clearly, we decouple the model learning component from the policy optimization component by testing the model on data coming from the replay buffer of a pre-trained model-free SAC agent. Such a setup ensures that the changes in state distribution according to changes in policy are still present but any changes

in the model do not actively affect this distribution, thus clearly reflecting the effect of model learning only.

We observe that our invariant model learner performs much better than the standard model learner, especially when the number of samples available is low, i.e., around the 200K to 500K mark (see Figure 4, left plot). As the number of samples increases, the performance between both models converges, just as observed in the tabular case. This is expected since in the infinite data regime, both solutions (MLE and invariance based) approach the true model. Furthermore, we observe that the number of samples it takes for convergence between the standard and the invariant model learners increases as the rollout horizon (H in Figure 4) of the model learner is increased. Next, we plot the test error when the invariance loss is computed for all state variables and compare it to when it computed for a particular state variable (Figure 4, middle plot). We see that the per-state version performs worse than the per-state-variable or per-dim version, showing the importance of enforcing invariance at the state variable level. Finally, we test the mean correlation in the dimensions of the learnt representation $\phi$ with and without model-invariance (Figure 4, right table). We see that the mean correlation is lower with model-invariance than without, suggesting that the practical algorithm does induce the sparsity property discussed in the earlier part of the paper.

| POPLIN | Cheetah | Walker | Hopper | Ant |
|---|---|---|---|---|
| *PETS | 2288 $\pm$ 510 | 282 $\pm$ 250 | 114 $\pm$ 311 | 1165 $\pm$ 113 |
| *POPLIN-A | 1562 $\pm$ 568 | -105 $\pm$ 125 | 202 $\pm$ 481 | 1148 $\pm$ 219 |
| *POPLIN-P | 4235 $\pm$ 566 | **597** $\pm$ 239 | **2055** $\pm$ 206 | 2330 $\pm$ 160 |
| *METRPO | 2283 $\pm$ 450 | -1609 $\pm$ 328 | 1272 $\pm$ 250 | 282 $\pm$ 9 |
| *SAC | 4035 $\pm$ 134 | -382 $\pm$ 424 | 2020 $\pm$ 346 | 836 $\pm$ 34 |
| SAC-SVG H-3 | 6530 $\pm$ 382 | 80 $\pm$ 472 | 1108 $\pm$ 263 | 2293 $\pm$ 397 |
| OURS H-3 | **7067** $\pm$ 269 | -150 $\pm$ 556 | **1724** $\pm$ 271 | **3124** $\pm$ 199 |

Table 2: **POPLIN Invariant MBRL performance** Invariant MBRL performance reported at 200k steps on four MuJoCo based domains from POPLIN [55]. * represents performance reported by POPLIN. Standard error with 10 seeds reported. We bold the scores with larger mean values.

| MBPO | Cheetah | Walker | Hopper | Ant | Humanoid |
|---|---|---|---|---|---|
| SAC-SVG H-3 | 8957 $\pm$ 532 | 3795 $\pm$ 503 | 3201 $\pm$ 101 | 3997 $\pm$ 153 | 1712 $\pm$ 415 |
| OURS H-3 | **9109** $\pm$ 334 | **3961** $\pm$ 239 | **3382** $\pm$ 84 | **4546** $\pm$ 286 | **2443** $\pm$ 561 |
| SAC-SVG H-4 | 8327 $\pm$ 870 | 3494 $\pm$ 392 | 3291 $\pm$ 232 | 4470 $\pm$ 307 | 2404 $\pm$ 495 |
| OURS H-4 | **9663** $\pm$ 487 | **4347** $\pm$ 136 | **3489** $\pm$ 19 | **4565** $\pm$ 221 | **3506** $\pm$ 538 |
| SAC-SVG H-5 | 5710 $\pm$ 1329 | 2773 $\pm$ 492 | 3059 $\pm$ 276 | 3808 $\pm$ 531 | 3190 $\pm$ 601 |
| OURS H-5 | **5796** $\pm$ 855 | **3326** $\pm$ 430 | **3207** $\pm$ 210 | **3817** $\pm$ 488 | **3446** $\pm$ 481 |

Table 3: **MBPO Invariant MBRL performance** Invariant MBRL performance reported at 200k steps on five MuJoCo based domains from MBPO [28]. Standard error with 10 seeds reported. We bold the scores with larger mean values.

**Invariant Model-based Reinforcement Learning**. Finally, we evaluate the invariant model learner within the policy optimization setting of SAC-SVG [2]. We compare the difference in performance to SAC-SVG when the horizon length is varied (see MBPO environments in Table 3 and Appendix 3.4) and then compare the performance of our method against multiple model based methods including PETS [11], POPLIN [55], METRPO [32], and the model free SAC [27] algorithm (see POPLIN environments in Table 2 and Appendix 3.4). The results in Table 3 show a consistent improvement in performance when the invariant model learner (OURS) is used instead of the standard model learner (SAC-SVG) across most tasks, including the Humanoid-v2. Furthermore, we see that as the horizon length is increased, the difference in performance between the invariant and standard learners increases for most tasks. A comparison of model training errors for both cases is provided in Table 5 in Appendix. We see a consistently higher training loss for the invariant learner, indicating less over fitting. Finally, in Table 2 we see that the invariant learner outperforms all six baselines on 3 out 4 tasks, with the exception of Walker, where most methods fail to reach a positive score. Combining our invariant model learner with other policy optimization algorithms is therefore a promising direction for future investigation.

# 8  RELATED WORK

Planning based on structural assumptions on the underlying MDP have been explored in significant detail in the past [7]. The most closely related setting is of factored MDPs, but approaches that build on the factored MDP assumption have predominantly also assumed a known graph structure for the transition factorization [31, 50, 42]. The factored MDPs literature has carried two distinct ideas all along, one of sparsity and the other of factorized transitions/rewards. This paper focuses predominantly on sparsity. We rely on factorization for some theoretical results but the practical algorithm does not require factorization to work. Sparsity leads to the abstraction viewpoint introduced in this work, which allows us to scale our formalism to practical methods. It allows us to connect the initial formalism to the concrete problem of neural network generalization (without an abstraction-based formalism, it does not make sense to use neural networks).

On a separate axis, papers that deal with learning the graph structure and then doing RL [50, 16, 26], do so by constructing close to optimal transition functions, and then planning. In our practical method, we focus on only avoiding spurious correlations, which is computationally more efficient since by the time an exact model of the MDP can be learnt (usually done by checking all possible parent set candidates), most model-free methods can just learn the reward function and perform much better. Degris et al. [13] learn decision tree based models online while we focus on learning NN based models. Extending such exact methods to NN models remains an open problem. Guestrin et al. [25] state a similar idea to this work but simply assume that one is given basis functions that have limited scope (model-invariant abstractions).

As mentioned, other forms of state abstraction such as bisimulation [37, 58] do not consider sparsity whereas model-invariance does. Therefore, both forms of abstraction are in fact orthogonal to each other, and therefore can be combined on top of one another. In similar essence, a lot of works have proposed *value-aware* model learning objectives, which only model the minimal information required to predict the value function of the task [20, 17, 12]. This is again a complimentary idea to model-invariance, since we consider states spaces that are already minimal w.r.t. the value function, i.e., no task irrelevant information is present.

# 9  CONCLUSION AND FUTURE DIRECTIONS

This paper introduced a new type of state abstraction for MBRL that exploits the inherent sparsity present in many complex tasks. We first showed that a representation that only depends on the causal parents of each state variable follows this definition and is provably optimal. Following, we introduced a novel approach for learning model-invariant abstractions in practice, which can be plugged in any given MBRL method. Experimental results show that this approach measurably improves the generalization ability of the learnt models. This stands as an important first step to building more advanced algorithms with improved generalization for systems that possess sparse dynamics.

In terms of future work, there remain multiple exciting directions and open questions. First, to enable model-invariance, we could also look at other kind of approaches proposed recently such as the AND mask [43]. The AND mask specifically requires the data to be separated into multiple environments, and thus naturally suits the offline RL setting. Second, moving to pixel based input, the representation learning task becomes two-fold, including learning to abstract away the irrelevant information present in the pixels and then learning a model-invariant representation. Third, note that our theoretical results do not involve an explicit dependence on a sparsity measure, for example, the maximum number of parents any state variable could have. Including such a dependence would ensure tighter bounds. Fourth, it is worth asking how such an explicit constraint on model-invariance can perform as a standalone representation learning objective, considering the strong progress made by self-supervised RL.

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
