# OpenReview forum: "Model-Invariant State Abstractions for Model-Based Reinforcement Learning"
_ICLR.cc/2022/Conference — ICLR 2022 Submitted_

### Official Review · Reviewer_5dqM · 2021-10-29

**Correctness:** 4
**Technical Novelty And Significance:** 3
**Empirical Novelty And Significance:** 3
**Recommendation:** 3
**Confidence:** 4

**Main Review:**

===== After rebuttal: the authors didn't address my concerns. I'll decrease my score accordingly. ====

This paper is well motivated. The model invariance structure indeed rises naturally in continuous control tasks as argued by the authors in Section 7. Identifying the direct causes using invariant causal prediction methods in reinforcement learning could potentially have a wide range of applications.

The paper is well written and easy to follow. The toy example in Section 5 illustrates how ICP helps model learning. Technical part of this paper is self-contained and well-explained.

On the plus side, this paper directly tackles the problem of model learning, which is a central question in model-based RL area. The model learning algorithm proposed by this paper reduces spurious correlation in mode learning and improves state-of-the-art results in low data regime. More importantly, the model learning algorithm is self-contained and can be plugged into almost any existing model-based RL algorithms, and achieves impressive performance improvement. The empirical evaluations are solid.

On the minus side, my main concerns are the following.
-	In the beginning of Section 3, the invariant abstraction \phi_i is defined w.r.t. a fixed index i. In this case, it’s expected that the abstraction can be much smaller than the state space. However, in definition 1, the abstraction \phi must hold for every index i (although w.r.t. different errors \epsilon_{i}). In this case, the existence of a small abstraction is unclear to me. For example, in the toy example in Section 5 (where there are three state variables and each depends only on its own), what’s the minimal model invariant abstraction according to definition 1? If I understand correctly, the transition probability must be the same for every index i, which implies that different states must have different abstraction.
-	It’s hard for me to follow the intuition behind Algorithm 2. In both the theory and the toy example, PA(x^i) is predicted using the invariance across different policies/environments. Is algorithm 2 using the same invariance? It seems to me that algorithm 2 regularizes the learned feature by requiring the similarity of separately learned transition dynamics. Why this regularization helps reducing spurious correlation (not only the performance. Although the improvement in performance is impressive, the core intuition introduced in previous sections is to reduce spurious correlation), in either theoretical or empirical perspective?

Additional questions.
-	What’s |\phi| in the paragraph below Theorem 1, Page 4? Is it referring to the number of abstract states or the side of hypothesis space of state abstraction?
-	The toy example focus on offline setting with replay buffers from multiple policies. But in the online setting, the policy change is expected, so naturally there will be multiple policies. What’s the reason for considering offline setting? Does the ICP method work for the online setting?
-	Table 1 shows that the transition prediction error is lower using model-invariance. But for Mujoco tasks, the error is higher. The authors claim that in both cases, model invariance helps model learning (but for different reasons). Why the results/arguments are not consistent?
-	What does “linearity in dynamics” mean in the bottom of Page 5? What’s the exact transition probability?
-	The variable k is not defined in Eq. (1).

Minor issues:
-	Broken refs: Line 3 of Alg. 1; below Theorem 1 in page 17; first line of Section 3.5 in Appendix, page 20.

Some suggestions that don’t related to my assessment:
-	All of the empirical results focus on low data regime (which, by the way, is perfectly fine because that’s the scope of this paper). It would be helpful to also include a result on the speed of convergence, and also the final performance upon converngence.
-	The inner product notation in the first line of page 7 is a little confusing, because standard inner products are commutative.


**Summary Of The Paper:**

This paper defines a novel model-invariant state abstraction for factored MDPs. It’s also shown that using invariant causal prediction significantly reduces transition prediction error in both toy example and several continuous control tasks. Inspired by theoretical results, this paper also proposes a novel method for learning model-invariant representations using neural networks. The model learning method can be combined with combined with almost any model-based deep RL algorithm. Empirically, this paper shows that in continuous control tasks, their model learning algorithm improves the performance of MBRL algorithms in the low data regime.

**Summary Of The Review:**

This paper is well-motivated, the empirical results are impressive. My main concerns are (1) the existence of small model-invariant abstraction, and (2) the intuition of Alg. 2. As a result, I can only recommend a weak accept at this point. But I’m open to raising my score if my concerns are addressed.

---

### Official Review · Reviewer_9B8W · 2021-11-01

**Correctness:** 3
**Technical Novelty And Significance:** 3
**Empirical Novelty And Significance:** 3
**Recommendation:** 5
**Confidence:** 4

**Main Review:**

In general, I think that the paper is well-written and I like the core idea of this paper because it points out an important direction to improve the data efficiency and generalization ability of the dynamics model. However, I cannot give the accept score in this procedure because there are some unclear points in this paper, but I am willing to increase my score if the authors fix the below problems in the rebuttal process.

1. The authors only show the effect of spurious correlation on the Humanoid env, which is not enough to validate that removing spurious correlation is important for RL problems. It is better to show more examples.

2. Similar to problem 1, the authors only report the mean correlation metric with and without the proposed model-invariance loss, can the authors provide more evidence to validate the effectiveness of model-invariance loss for removing spurious correlation?

3. Why do not explicitly learn the causal structure of the dynamics? What is the advantage of the proposed implicit model-invariant loss compared with explicitly learning and exploiting the causal structure?

4. Would the performance of the proposed invariance loss be improved if we increase the number of randomly initialized dynamics models?

5. I notice that the difference in performance between the invariant and standard learners reduces when the horizon increases from 3 to 5 for most tasks, which violates the claim
>the horizon length is increased, the difference in performance between the invariant and standard learners increases for most tasks.

 Why is this a phenomenon? Can authors add an explanation for it?

6. The authors only show the training env prediction errors in Table 5, but do not report test env prediction errors, which is insufficient to support the claim
>We see a consistently higher training loss for the invariant learner, indicating less overfitting.

Minor Points:
There are several missing Table/Section references, e.g. Algorithm 1/ Appendix 3.5.


**Summary Of The Paper:**

This paper points out that the causal structure is important for the generalization ability of a dynamics prediction model, and empirically show that learning a model-invariance representation can implicitly learn the causal sparsity and improve the generalization ability of the learned model.

**Summary Of The Review:**

In summary, the paper focus on an important topic of the reinforcement learning area. The paper is also well-written, but the analysis and the experimental support are not strong enough, so I give 5 in this procedure, but I am willing to increase my score if the authors fix them in the rebuttal process.

---

### Official Review · Reviewer_hNYg · 2021-11-02

**Correctness:** 2
**Technical Novelty And Significance:** 2
**Empirical Novelty And Significance:** 2
**Recommendation:** 3
**Confidence:** 3

**Main Review:**

The paper presents in interesting, useful idea that causality plays in important role in generalization performance in model-based reinforcement learning, and proposes an interesting method to leverage this. The paper rests on a fundamental assumption that in many cases, the full state of a system can be decomposed, with the probabilistic model governing each state dimension being independent given the previous state and action, and furthermore, each state dimension being causally dependent only on a select portions of state vector. While this isn't true generally, I buy the argument that for a lot of systems with high dimensional state representations, such as multi-agent systems or the MuJoCo systems considered in this work, this may be approximately correct. Indeed, even if this isn't exactly true, it may be worth making this assumption for improved generalization performance in the early stages of MBRL where data is sparse.

My main concerns with this work are the following:

- Disconnect from theory to practice:
    - The theoretical insights connecting RL to ICP are only valid for the very limited linear MDP setting. While in many cases, it is useful to derive theory in a linear case to inform practical algorithms for the nonlinear case, but in this work, the practical approach put forth in the paper seems disconnected from the linear case. There is fundamentally different strategy for intervention, moving from interventions on the data to interventions in the form of learning models from different data samples. There is no call to a ICP subroutine; instead, the approach adapts the ReLIC architecture for self-supervised intervention-driven representation learning. While the connections to causal prediction are clear for the ReLIC model, the architecture proposed by the authors is different, and indeed designed for a supervised learning problem rather than the self-supervised learning problem tackled by ReLIC. These differences make it hard for me to connect the practical algorithm proposed by the authors to the causal learning ideas they use to motivate it.

- Lack of clarity:
    - I found several details to be hard to follow in the presentation of this work. For example, in Lemma 1, $P_\phi(s,a)$ is defined to be a mapping from $s,a$ to a distribution on $\mathcal{X}$, as it is the expectation over $P(\cdot | x, a)$, which is a distribution on $\mathcal{X}$. However, in the text, it is described as a transition model in the lifted state, which to me implies it should be a mapping from $s,a$ to a distribution on the abstract state space. Furthermore, in the nonlinear architecture, the details of the invariance loss are confusing. The text refers to $\psi$ as the cosine similarity between the predictions of $f$ and $h$, but the definition suggests an inner product between the projections of the predictions. The authors should be more clear if they are using the angled brackets to denote cosine similarity rather than the standard inner production. More importantly, the definition of $\psi^i(f,h)$ as an inner product suggests it maps to a scalar, so it doesn't make sense to apply a softmax to this result directly. The pytorch-like code in the appendix does not provide any additional clarity, as the code does does not distinguish between elementwise and matrix multiplication and the dimensionality of the variables is not clear.
    By referring to the ReLIC paper, I was able to determine that the softmax is applied over the batch dimension, implying a auxiliary task of instance identification within the batch, and a desire to enforce invariance to interventions for this task. However, this does not make sense for the approach in this paper, since model learning is a supervised task, so one could enforce invariance directly on the predictions of the next state. Indeed, in the ReLIC paper, the loss includes both a invariance term computed on $p^{fh}$, and a instance identification loss on $p^{fh}$. However, in the proposed approach, there is only a KL loss, so what is stopping the projection network $g$ from learning a trivial projection mapping all inputs to the same vector, such that  $p^{fh} =p^{hf}$ without enforcing any invariance on the actual predictions of the next state?

Minor comments / questions:
- Do you have references to support why Assumption 2 is reasonable? What is the distinction between $x_{t}$ and x_{t+1}$ over a rollout of $x_1, a_1, \dots, x_T$.
- What, specifically, was the linear MDP used in the tabular RL experiment? The specific choices of MDP and policies used to create ICP environments should be detailed in the appendix.
- Bolding results based purely on mean value is nonstandard and obscures the fact that in many cases the proposed approach is not statistically significantly more performant than the SAC-SVG baseline, e.g. in table 3. Additionally, why is the result for "Ours H-3" on Hopper in table 2 bolded?
- In Figure 4, the test model error does not seem to have converged by the end of the plot. It would be useful to see this asymptotic performance to show whether the invariant model converges to the same quality of model as the standard model.

**Summary Of The Paper:**

This work presents a method for learning representations for model based reinforcement learning by leveraging assumed sparsity in the causal graph to improve the generalization performance of the learned model in the low-data regime. The authors propose a method which, at a high level, aims to utilize data collected under different conditions (e.g. policies) to extract a representation under which the transition model is approximately invariant. Furthermore, they provide a bound on difference between the optimal value function on the learned representation and the true optimal value function as a function of the approximation quality of the invariant model. They show how invariant causal prediction can be used in the the linear tabular setting to yield improved performance relative to learning a model directly in the original state space. Next, they present an approach to achieve a similar objective for the nonlinear case, presenting a neural network architecture based on that proposed in ReLIC, and show that the proposed architecture improves the low-data performance in learning dynamics models, and leads to improved performance relative to baselines.

**Summary Of The Review:**

Overall, I found that this paper has a good, theoretically motivated derivation to improve low-data generalization in MBRL, which is a clear bottleneck. However, the proposed practical algorithm was hard to understand, and fundamentally disconnected from the theoretical analysis presented in this paper. Hence, I do not support acceptance as is. In a revised version of the paper, I recommend that the authors draw stronger connections between causal prediction to the practical implementation that they propose.

---

### Official Review · Reviewer_avfm · 2021-11-03

**Correctness:** 3
**Technical Novelty And Significance:** 2
**Empirical Novelty And Significance:** 2
**Recommendation:** 3
**Confidence:** 4

**Main Review:**

- The idea of combining the two concepts of sparsity and causal invariance is not new and has been frequently mentioned in the causality literature, e.g., sparse shift mechanism etc. See more in [1].

- In Assumption 1, the authors do not consider instantaneous effects, which render such an assumption quite limited in real world applications.

- In the paragraph of **Invariant Causal Prediction** (Section 2), the authors describe ICP as learning an invariant representation. This is misunderstanding about ICP, because the representation is allowed to vary across environments. In fact, in ICP, it is only assumed that the causal mechanism is invariant, instead of the representation.

 - In Definition 1, "$:=$" under $\sup$ should be corrected.

- It is NOT true to say that "This change in distribution (over states and action) can be realized as a change in the policy". Since $p(x, a)=p(a|x)p(x)$, the change in $p(x, a)$ can result only from a change in $p(x)$, when the policy $p(a|x)$ is invariant. Thus, the subsequent claim "each policy \pi defines an ICP environment $e$" is also wrong. All this might lead to some inconsistencies in the rest of the paper.

- I also have some doubts about the claim that "when sparsity is present, following the above definition leads to a representation $\phi$ such that any two components $\phi_i$ and $\phi_j$ are decorrelated." Because $\phi_i$ and $\phi_j$ could have some overlap, in this case the two components are no longer decorrelated.

- I feel that the theoretical results presented in the paper are not quite new, all of which are immediate results of or simple extension to previous work. Specifically, Proposition 1 is just an immediate result of directly applying ICP to the MDP setting. Theorem 1 is also a simple extension to Theorems 2&3 in [2].

- Missing appendix index in Algorithm 1.

- How to select the model $h$? Why are the parameters of $h$ not updated during the training phase? How to determine the number of models? Because two models might not be enough in many scenarios.

- In the linear settings, it is convincing that the learned model could generalize well to unseen environments, because causal variables can be identified by means of ICP. However, this is not the case in the nonlinear settings. As we know, invariance does not imply causation. Hence, the proposed approach (i.e., Eq 2) to approximately learning such abstract states has no guarantees on that causal variables can be identified from the training data. This leads to no guarantees on good generalization performance in unseen environments.

- All the baselines to which the proposed approach is compared are not quite related to invariant predictors or something similar that this paper studies. This makes it hard to judge the validity of the proposed approach. I suggest that the authors should compare it with more relevant methods. For example, a natural and simple baseline to compare with is IRM, where IRM can be directly used to learn the invariant transition function.

References:

[1] Schölkopf et al. Towards causal representation learning. 2021.

[2] Zhang et al. Invariant causal prediction for block MDPs. 2020.

**Summary Of The Paper:**

The authors introduce a new type of state abstraction called model-invariance by integrating sparsity with causal invariance in the single-task RL scenarios. In the linear settings, they learn the model-invariance states by directly applying ICP and prove that an optimal policy can be learned over them. In the nonlinear settings, they instead propose a practical method to approximately learn such model-invariance states.

**Summary Of The Review:**

On the theoretical side, the main theorems are either immediate results of or simple extension to the previous work. On the technical side, they draw on ICP and ReLIC in the linear and nonlinear settings, respectively. On the experimental side, the baselines are not convincing enough to judge the validity of the proposed approach.

---

### Decision · Program_Chairs · 2022-01-20

**Decision:**

Reject

**Comment:**

This paper proposes a method to learn representations in MBRL by exploiting sparsity in the model to improve data efficiency. The key idea is to build a representation for which the model is invariant.
The idea is quite interesting, but one weakness of the current draft is that there is a disconnect between the presented theory (linear case) and the relevant experimental setup (non-linear).
The paper is overall well written but would still benefit from a revision to improve clarity as pointed out by the reviewers.
The experimental results are inconclusive due to the choice of weak baselines.